# A study on altruistic consumption of Chinese residents from the perspective of intergenerational income mobility

Yingying Shao⦿*

School of Economics, Shandong University of Finance and Economics, Jinan, Shandong, China

* susie2000@126.com

**Data Availability Statement:** The data used in my manuscript can be found in the following database: China Family Panel Studies (CFPS) https://opendata.pku.edu.cn/dataverse/CFPS.

## Abstract

Resident consumption is an important link to smooth the domestic circulation and promote the economy to achieve high-quality development. To stimulate the vitality of residents' consumption and achieve the expansion and quality of consumption, we should not only focus on the scale and willingness of residents to consume, but also consider the motivation of consumption. The paper analyzes the impact of intergenerational income mobility on residents' marginal propensity to consume and consumption altruism motive by expanding the Over Lapping Generation Models and combining the China Family Panel Studies (CFPS) data. The results of the study show: Intergenerational Income Elasticity has a significant positive effect on residents' marginal propensity to consume. When Intergenerational Income Elasticity increases, the "altruistic motive" of residents' consumption will also increase significantly. Compared with rural residents, Intergenerational Income Elasticity has a stronger positive effect on urban residents' marginal propensity to consume and marginal propensity to consume education. With the effect of intergenerational factors, urban residents' consumption shows stronger altruistic motives.

## 1 Introduction

For more than 40 years of reform and opening-up, China has established a socialist market economy system in line with its own national conditions, which has stimulated economic and social vitality and created a miracle of economic growth. As an important component of total social demand, the role of consumption in China's national economy has become increasingly prominent. According to the data published by the National Bureau of Statistics of China, in 2014, China's total final consumption was 338,031.2 billion yuan. And this indicator reaches 619,688 billion yuan in 2021. From 2014–2021, the contribution of final consumption to economic growth topped the three major demands in all years except 2020. In China, about 70% of final consumption comes from residential consumption. At constant prices, the average annual growth rate of China's residential consumption during 2014–2021 is 7.29%, which is significantly higher than the average annual growth rate of GDP during the same period. Consumption, as the final demand, is a direct manifestation of people's aspiration for a better life. Enhancing the fundamental role of residential consumption on economic development is the

**Funding:** The author(s) received no specific funding for this work.

**Competing interests:** The authors have declared that no competing interests exist.

key to building a domestic and international double-cycle development pattern and achieving high-quality economic development.

Although the scale of China's residential consumption is rising year by year in absolute terms, the release of China's residential consumption potential is not sufficient. The problem of insufficient relative amount of consumption and low consumption rate is still prominent. Data from the National Bureau of Statistics of China show that since 2014, the consumption rate of our residents is still less than 40%, although it has increased compared to the past. The consumption rate of our residents were 39.2%, 38.2% and 38.4% in 2019, 2020 and 2021. The consumption rate of our residents is not only lower than that of developed countries in the same period, but there is also a gap even when compared with countries and regions at a similar stage of economic development. According to the data published by the World Bank, in 2019–2021, the consumption rates of residents of four BRICS countries, Brazil, Russia, India and South Africa, are more than 50%, of which the consumption rates of residents of South Africa and Brazil are more than 20 percentage points higher than that of China.

To analyze the current consumption situation of Chinese residents and explore the root causes of the low consumption rate, we should not only follow the general rules of consumption theory, but also take into account the specific reality of Chinese residents. Influenced by the traditional Chinese culture of "family concept", families in China show a strong intergenerational bonding. This intergenerational bond has been further strengthened by the development of the national economy, the rising income levels of the population, and the gradual increase in life expectancy. The "altruistic motive" of residential consumption is magnified by intergenerational factors, and its influence on residents' consumption decisions is becoming more and more prominent. It is important to examine the impact of intergenerational income mobility on residents' altruistic motives of consumption from an intergenerational perspective, and to explore effective solutions for releasing residents' consumption potential and promoting consumption upgrading to enhance the overall effectiveness of the national economy.

The paper is structured as follows:

Part 1 introduces the background of the paper and the significance of the study. Part 2 introduces Diamond's Overlapping Generations Model. This part also provides an overview of the research literature on intergenerational income mobility around the topic of this paper. Part 3 constructs a theoretical model of how the level of intergenerational income mobility affects the marginal propensity to consume and the altruistic motivation to consume in residential households by extending the Overlapping Generations Model. Part 4 describes the model, variables and data for the empirical analysis of this paper. Part 5 presents the results of the empirical analyses. Firstly, it shows the results of the Intergenerational Income Elasticity (IGE) for China over the period 2010–2018. Secondly, it shows the estimation results of the IGE affecting residents' marginal propensity to consume and marginal propensity to consume education in China. Heterogeneity analysis is also done from an urban-rural perspective. Part 6 presents the conclusions of the study and related recommendations in terms of increasing the level of intergenerational income mobility and diminishing the altruistic motive for consumption.

## 2 Related literature

### 2.1 Diamond's overlapping generation model

The concept of overlapping generation was first proposed by Samuelson in 1958, and Diamond built on it in 1965 by proposing a more systematic model: Overlapping Generations

model (OLG) [1]. The OLG model takes into account the overlap between the old and the new in the population and divides the human life into two periods: youth and old age. Individuals earn income through labor in youth and allocate their income between consumption and savings in youth. Individuals have no income from labor in old age, and consumption expenditures come from savings and their interest income in youth. Therefore, society has two generations simultaneously in period $t$: young people born in period $t$, and older people born in period $t-1$ who have entered old age. When the older generation passes away, the former young people enter old age and a new population becomes young.

The model allows for variability in consumption behavior across generations and sets up a utility function for consumers. The model constrains the level of income, sets the rate of technological progress as an exogenous variable, and examines individual consumption decisions by solving for the maximum of an economic individual's lifetime intertemporal consumption utility.

## 2.2 Literature on intergenerational income mobility

In the field of intergenerational income mobility research, a number of literatures have focused on the measurement of the level of intergenerational income mobility and the exploration of its influencing factors (Kamhon K et al., 2014; Wang et al., 2019; Liu and Zhao, 2020) [2–4]. It is worth paying attention to the fact that although the intergenerational transmission of wealth does not change the current total social wealth, it can affect the social income gap and the level of human capital, thus becoming an important entry point for some scholars to analyze residents' consumption behavior and consumption level. Scholars generally agree that there is an inverse relationship between the level of intergenerational income mobility and income disparity. Literature (Maoz and Moav, 1999) suggests that high levels of intergenerational income mobility coexist with low income gaps in societies with high levels of economic development [5]. The cost of education is the main determinant of the inverse relationship between income disparity and intergenerational income mobility (Nakamura and Murayama, 2011) [6]. Literature (Corak M, 2013) uses the Gini coefficient as the horizontal axis and the Intergenerational Income Elasticity (IGE) as the vertical axis, and places multi-country data in the coordinate system to obtain the "Great Gatsby Curve", concluding that the stronger the intergenerational income mobility, the smaller the social income gap [7]. Corak's findings were verified by using data from different countries (Guido, 2016; Yang and Liu, 2019) [8, 9].

Regarding the impact of children's education expenditures on household consumption, two distinct views have been formed. Some scholars believe that an increase in children's education expenditures will cause a decrease in household consumption levels of residents. At a given income, increased investment in children's education can have a significant crowding-out effect on other household consumption, thus inhibiting the improvement of household living standards (Yang and Chen, 2009; Long and Liang, 2019; Yang and Zhang, 2019) [10–12]. From a precautionary savings perspective, spending uncertainty is a key motivation for households to save precautiously (Li et al., 2018) [13]. Parents with altruistic motives choose to increase savings and reduce consumption in order to cope with the uncertain risk of education spending (Yuan et al., 2014; Ji and Ning, 2020; Xu et al., 2021) [14–16]. Influenced by traditional beliefs, Chinese households are less willing to smooth consumption through debt, so the uncertainty of education spending will have a greater impact on Chinese households' precautionary savings (Zhu and Xia, 2018) [17]. There is also a view based on the transmission path of "education-human capital-income consumption", which suggests that education expenditure has certain investment properties and can increase labor force income through human capital accumulation, which in turn can further increase household consumption (Liu et al.,

2021) [18]. In addition, education can also influence people's consumption perceptions, consumption habits and consumption preferences, contributing to the improvement of household consumption propensity and the optimization of consumption structure (Di et al., 2019) [19].

This paper extends the OLG model to analyze the intrinsic association between intergenerational income mobility and residents' propensity to consume, and constructs a theoretical framework in which intergenerational income mobility affects residents' marginal propensity to consume and altruistic motives, further enriching consumption theory. The paper examines the effect of IGE on residents' marginal propensity to consume and altruistic consumption using data from China Family Panel Studies (CFPS) (2010–2018), and measures the contribution of Intergenerational Income Elasticity using a Shapley value decomposition. This paper examines the heterogeneity of the role of Intergenerational Income Elasticity on residents' altruistic consumption motives from urban and rural perspectives, compares the differences in the effects of Intergenerational Income Elasticity on consumption motives of different groups, and contributes more ideas to realize the quality and expansion of residents' consumption and overall economic efficiency in China.

## 3 Methodology

Based on Diamond's Overlapping Generation Model (OLG), the paper divides the survival period of individuals into adolescence, middle age and old age. The thesis has the following assumptions: Individuals are not able to work during adolescence and need to be raised by their parents, while adolescence is also the main stage in which individuals accumulate human capital through education. In middle age, the individual works and earns personal income through labor, raises children, pays for their education, and may also transfer wealth (out or in) with his or her parents during this period. In old age, the individual is not able to work and therefore cannot continue to earn income from labor, and the living expenses are determined by a combination of savings in middle age and intergenerational wealth transfers. Since the paternal generation has already died out, there is the possibility of wealth transfer between individuals and their offspring only at this period. It is also assumed that intergenerational maintenance and wealth transfers occur only between neighboring generations.

The model uses 1, 2, and 3 to denote adolescence, middle age, and old age, respectively. Adolescence are defined as those who are not yet able to work and are in the education stage. Middle aged residents are defined as those who are able to earn income from labor and need to raise children. And old aged residents are defined as those who are not able to work and therefore cannot continue to earn income from labor.

Denote the economic individual by $t$, its parent by $t$-1, and its offspring by $t$+1. Combined with altruistic motives, the individual's utility function can be expressed in the following form:

$$U_t = \ln C_{2,t} + \theta_1 \ln C_{3,t} + \theta_2 \cdot n \ln Y_{2,t+1} + \gamma_1 \ln C_{3,t-1} \tag{1}$$

In the above equation, $C_{2,t}$ and $C_{3,t}$ denote individuals' consumption levels in middle and old age, respectively; $Y_{2,t+1}$ denotes their individual children's income levels in middle age; $C_{3,t-1}$ denotes their fathers' consumption levels in old age; $\theta_1$ is the time discount factor, $\theta_2$ and $\gamma_1$ denote individuals' altruistic tendencies toward their children and fathers, respectively, and $n$ is the number of children.

Each individual can earn income through work only in the middle age. Therefore, from one's own perspective, the individual's income in middle age has to pay for current consumption in addition to saving to meet consumption expenditures in old age. From an intergenerational perspective, the income in middle age has to cover the education expenses of children

and there is a possibility of wealth transfer with their paternal generation.

$$C_{2,t} = Y_{2,t} - nl_{1,t+1} + B_{3,t-1} - S_{2,t} \tag{2}$$

In Eq (2), $C_{2,t}$ and $Y_{2,t}$ denote the consumption expenditure and income level of economic individuals in the middle age, respectively; $I_{1,t+1}$ denotes the education expenditure of economic individuals for a single minor child; $S_{2,t}$ denotes the savings of economic individuals in the middle age; $B_{3,t-1}$ denotes the wealth transfer from the father to the individual, and $B_{3,t-1}>0$ indicates that there is wealth transfer from the father to the economic individual, and $B_{3,t-1}<0$ indicates that there is wealth transfer from the economic individual to the father.

The consumption of the individual in old age depends mainly on two aspects: first, the savings and interest income in mid-life, and second, the amount of wealth transferred to children in old age. It is expressed as a function of:

$$C_{3,t} = (1 + r) \cdot S_{2,t} - nB_{3,t} \tag{3}$$

In the above equation, $C_{3,t}$ denotes the consumption expenditure of the individual in old age and $r$ is the interest rate.

Factors such as the individual's expenditure on the education of the offspring, the amount of wealth transfer, and the offspring's individual talent and opportunity determine the offspring's income level in middle age. The Cobb-Douglas function form is used to express the income of the offspring in the middle age.

$$Y_{2,t+1} = (\alpha_1 I_{1,t+1} + \alpha_2)^\beta (B_{3,t} + H)^{1-\beta}, \beta \in (0, 1) \tag{4}$$

In Eq (4), $\alpha_1$ is the return to education, $I_{1,t+1}$ denotes the education expenditure of economic individuals on a single minor child, $\alpha_2$ denotes the individual talent level of the offspring, $B_{3,t}$ denotes the amount of wealth transfer received by the offspring from its parent, $H$ is the amount of capital from external factors such as chance and luck. $\alpha_1 > 0, H > 0$.

The level of consumption of an individual's paternal generation in old age depends on the accumulation of wealth in the middle age of the paternal generation and the amount of wealth transferred between the individual and the paternal generation.

$$C_{3,t-1} = W_{2,t-1} - B_{3,t-1} \tag{5}$$

In Eq (5), $C_{3,t-1}$ denotes the consumption expenditure of the economic individual's parent in old age, $B_{3,t-1}$ denotes the amount of wealth transfer from the parent to the individual, and $W_{2,t-1}$ denotes the amount of wealth accumulation of the parent in middle age, which is considered as an exogenous variable.

Using Eqs (2)–(5) as constraints, find the equilibrium solution when individual utility is maximized.

$$I_{1,t+1}^{(*)} = \frac{\theta_2 \beta}{1 + \theta_1 + n\beta\theta_2} \left[ Y_{2,t} + B_{3,t-1} - \frac{n}{1+r} B_{3,t} - \frac{\alpha_2(1+\theta_1)}{\alpha_1 \theta_2 \beta} \right] \tag{6}$$

$$C_{2,t}^{(*)} = \frac{1}{1 + \theta_1 + n\beta\theta_2} \left( Y_{2,t} + B_{3,t-1} + \frac{n\theta_1}{1+r} B_{3,t} + \frac{n\alpha_2}{\alpha_1} \right) \tag{7}$$

From Eqs (6) and (7), the paper derives the individual's marginal propensity to consume at equilibrium.

This paper uses the Intergenerational Income Elasticity (IGE) to measure the strength of the level of intergenerational income mobility. According to Solon's (1992) definition [20], the

a paper expresses the IGE as:

$$IGE = \frac{\partial \ln Y_{2,t+1}}{\partial \ln Y_{2,t}} \tag{8}$$

From Eq (8), it can be seen that the IGE measures the degree of correlation between the income of the offspring and the income of the father. The larger the result of the IGE calculation, the more prominent the dependence of the income of the offspring on the income of the father and the weaker the intergenerational income mobility.

Combines Eqs (4), (6) and (8) and get the following results:

$$IGE = \frac{\partial \ln Y_{2,t+1}}{\partial \ln Y_{2,t}} = \frac{(1+r)\alpha_1 \beta Y_{2,t}}{(1+r)\alpha_1(Y_{2,t} + B_{3,t-1}) - n\alpha_1 B_{3,t} + n\alpha_2(1+r)} \tag{9}$$

$$\frac{\partial c_1^*}{\partial IGE} = \frac{n\theta_1 \theta_2 [\alpha_1(1+r)(Y_{2,t} + B_{3,t-1}) - \alpha_1 n B_{3,t} + n\alpha_2(1+r)]}{\alpha_1(1+r)Y_{2,t}(1 + \theta_1 + n\beta\theta_2)^2} \tag{10}$$

From Eqs (2) and (3), it can be derived that $Y_{2,t} + B_{3,t-1} > S_{2,t}$, and $(1+r) S_{2,t} > nB_{3,t}$. Therefore, $(1+r)(Y_{2,t} + B_{3,t-1}) > nB_{3,t}$, and the calculation of Eq (10) is positive. This implies that there is an isotropic relationship between IGE and the marginal propensity to consume of the paternal generation.

The intergenerational economic association between the offspring and the father in the theoretical model is manifested in two main ways: direct transfer of wealth and parental investment in the education of the offspring. Direct transfer of wealth is mainly in the form of money or valuables. If the direct transfer of wealth can generate income for the offspring, such income is mainly in the form of property income. The father's investment in the offspring's education is reflected in the offspring's human capital, and the level of human capital directly determines the level of wage income and business income. According to the data published by the National Bureau of Statistics of China, in the last decade, about 7%-9% of the income composition of China's residents is property income, and over 70% is wage and business income. This means that for the vast majority of families, direct wealth transfers alone make it difficult for the offspring to maintain the same income bracket as the father's generation. Educational inputs from the father's generation will play a more significant role in the income of the offspring.

When the IGE increases, it becomes more likely that the offspring will be in the same income bracket as the father. Assume that the father has the ability to accumulate wealth for the offspring and that the father has a given income expectation for the offspring. In order to help the offspring reach the expected income level and maximize their own utility, the father needs to make a trade-off between "direct wealth transfer" and "spending on the offspring's education". Based on the realistic income structure, the father's generation will be more inclined to reduce the size of direct wealth transfer and increase the expenditure on education for the offspring. This choice will lead to lower savings and higher consumption expenditures by the father's generation, and the father's marginal propensity to consume will increase. In order to increase the income level of the offspring, the share of education spending will be more pronounced in the increased consumption propensity of the father's generation. The increase in Intergenerational Income Elasticity leads the father's generation to show stronger altruistic motives in consumption.

It is further important to note that the size of the intergenerational transfer of wealth is based on the saving capacity of the paternal generation, which in turn is directly determined

by the level of income. Residents in the lower income bracket have less ability to save and leave less direct wealth transfers to their offspring, and less ability to "reduce the size of wealth transfers to increase spending on their offspring's education". Therefore, if the father's generation wants to boost the offspring's intergenerational leap by investing more in education, a significant portion of the investment in education will be achieved by cutting back on other household consumption. This restructuring within consumption does not significantly change the consumption propensity of the resident. Moreover, due to the low consumption base of the low-income group and the existence of spontaneous consumption, there is limited room to increase spending on children's education by crowding out other consumption, resulting in less altruism in the consumption of the low-income group.

Based on the above analysis, the following hypotheses can be made:

*H1*: IGE is positively related to the marginal propensity to consume of the residents.

*H2*: IGE is positively correlated with the share of education expenditure in residents' consumption. The higher the Intergenerational Income Elasticity, the stronger the altruistic motive of residents' consumption.

Combined with the fact that there is a long-standing urban-rural income gap in China, it is further speculated that:

*H3*: The positive effect of IGE on the marginal propensity to consume of urban residents is more prominent.

*H4*: Based on intergenerational considerations, the altruistic motives of urban residents' consumption are more pronounced than those of rural residents.

## 4 Variables, models and data

### 4.1 Variables

**4.1.1 Core explanatory variable.**   The core explanatory variable in this paper is the Intergenerational Income Elasticity (IGE). Most of the existing literatures on measuring IGEs use Solon's (1992) model design in which the age of the offspring, the age of the father and their squared values are included in a log-linear model of the offspring's income and the father's income to control for the effect of the age factor on the estimation results [20].

$$\ln Y_{i1} = b_0 + b_1 \ln Y_{i0} + b_2 age_{i1} + b_3 age_{i1}^2 + b_4 age_{i0} + b_5 age_{i0}^2 + \varepsilon_i \qquad (11)$$

In Eq. (11), $Y_{i1}$ and $Y_{i0}$ represent the income of the offspring and the income of the father of family *i*, respectively. $age_{i1}$ and $age_{i0}$ denote the ages of the offspring and the father. The estimated value of $b_1$ is the IGE, $b_2$, $b_3$, $b_4$, $b_5$ are the coefficients of the control variables, and $\varepsilon_i$ is the error term. Strictly speaking, the income levels of offspring and the income of the father in Eq (11) should be persistent income, but persistent income presents practical difficulties in data acquisition. To enable more accurate estimation of IGE, scholars have treated income levels using different methods. Some scholars use the average income method, which uses the average of the income of individuals over many years in place of persistent income (Mazumder, 2005) [21]. Some scholars have also used single-year income data for estimation, along with an instrumental variables approach (Cavaglia, 2015; Yang and Shi, 2016) [22, 23] or restrictions on the age of the offspring (Xu et al., 2021) [16] to overcome the problem of possible estimation bias due to single-year income. Some other scholars use dual measures to estimate IGE (Yang and Wang, 2020) [24]. The CFPS database used in the paper does not have

exactly the same individuals surveyed in each year, and it is not appropriate to use the average of individual multi-year incomes as a proxy for persistent income levels. This paper chooses to use individual single-year income to represent their persistent income and uses the instrumental variables approach to estimate the model. In addition, in the current society, for a large proportion of families, the mother's income from work is also an important component of the family income, so this paper uses "the sum of parents' income" to represent the income of the father.

**4.1.2 Explained variables.** The explained variable of this paper is the Marginal Propensity to Consume (MPC), which also needs to be estimated. Based on Keynes' Absolute Income Hypothesis, a linear model of consumer spending and income can be used to measure the Marginal Propensity to Consume overall or in groups.

$$C_i = a_0 + a_1 Y_i + a_2 X_i + \delta_i \tag{12}$$

In Eq (12), $C_i$ denotes the consumption expenditure of family $i$, $Y_i$ denotes the income level of this family. $X_i$ is a set of control variables, mainly including factors such as age of household members (age) and net household assets (asset). The estimated value of $a_1$ is the Marginal Propensity to Consume (MPC), $a_2$ is the coefficients of the control variables, and $\delta_i$ represents the error term.

The altruistic motive of consumption mainly describes the extent to which the consumer goods and services purchased by residents are intended to serve others. Most of the education expenditures of Chinese households are paid for their children, so there is a clear altruistic motive for this part of consumption. Combining the research logic of this paper, we will measure the Marginal Propensity to Consume from two perspectives. If $C_i$ in Eq (12) uses "all consumption expenditure", then the estimated MPC covers all types of consumption; if $C_i$ uses "education expenditure in consumption", then the marginal propensity to consume in education is estimated. These two types of marginal propensity to consume are denoted by $MPC_1$ and $MPC_2$, respectively.

**4.1.3 Control variables.** Referring to the relevant literature, the following control variables are used in this paper: (1) household income ($Y$); (2) net household property (*asset*); (3) income growth rate ($y$), the growth level of residents' per capita disposable income; (4) income gap (*gini*), using the Gini coefficient to reflect the income gap; (5) age (*age*); (6) consumption mode (*pattern*). The consumption pattern mainly reflects the proportion of residents' online consumption. Since the level of online shopping is positively correlated with the length of time spent online, this paper uses the "daily spare time of residents online" to reflect their consumption patterns; (7) education level (*edu*), measured by the number of years of education of an individual; (8) urbanization rate (*urb*), measured by the share of urban population in the total population; (9) population dependency ratio (*popu*), calculated using the "ratio of the sum of the juvenile and elderly populations to the working-age population", where the age of the juvenile population is defined as 0–14 years old and the age of the elderly population is defined as 65 years old and above.

## 4.2 Models

Based on the measurement of IGE and MPC, the following model is constructed to analyze the effect of Intergenerational Income Elasticity on residents' Marginal Propensity to Consume using provincial panel data.

$$MPC_{ct} = d_1 + d_2 IGE_{ct} + d_3 X_{ct} + \mu_{ct} \tag{13}$$

In Eq (13), the core explanatory variable $IGE_{ct}$ denotes the Intergenerational Income

Elasticity of province c in period t, the explained variable $MPC_{ct}$ denotes the Marginal Propensity to Consume of residents in province $c$ in period $t$, $X_{ct}$ is the set of control variables chosen above, and $\mu_{ct}$ represents the error term. $d_2$ is the coefficient of the effect of Intergenerational Income Elasticity on the Marginal Propensity to Consume of residents.

## 4.3 Data

First, the paper uses the China Family Panel Studies (CFPS) data to measure the intergenerational income mobility of Chinese residents in different years. We used person ID, family ID, father ID, and mother ID to match family data, person data, and family relationship data from CFPS in 2010, 2012, 2014, 2016, and 2018. A total of 10,356 sets of samples were obtained after removing samples with missing data or invalid data.

Second, the samples are grouped according to province ID and year. The IGE, $MPC_1$ and $MPC_2$ are measured for each group using Eqs (11) and (12). Individual characteristic variables such as household income ($Y$), net household property ($asset$), consumption mode ($pattern$), and education level ($edu$) among the control variables were calculated using the average of the values of corresponding variables in each group. Macro indicators such as Gini coefficient, income growth rate ($y$), urbanization rate ($urb$), and population dependency ratio ($popu$) use provincial data which are obtained from the statistical yearbooks of Chinese provinces and the website of the National Bureau of Statistics of China. After data processing and matching, 125 sets of provincial panel data were finally obtained.

# 5 Results

## 5.1 Intergenerational income mobility in China

Using collated mixed cross-sectional data, we estimate the IGE for China over the period 2010–2018. To address the downward bias and possible endogeneity problems caused by single-year income instead of persistent income in the model estimation process, we also estimate IGE using the parent's education level ($edu_0$) and International Socio-economic Index ($ISEI_0$) as instruments using Two-stage Least Squares estimation (2SLS). The results are shown in Table 1.

Because White's test and Breusch-Pagan test results showed that the model still had heteroskedasticity, we used robust standard errors for estimation. And column (1) in Table 1 shows the results. Column (2) shows the results of 2SLS estimation and instruments validity tests. The Cragg-Donald Wald F statistic is 762.702, and the p-value of the Sargan statistic is greater than 0.05. Therefore, it can be judged that there is no overidentification and weak identification of the instruments. $ISEI_0$ and $edu_0$ are valid instruments.

Both OLS and 2SLS estimation results in Table 1 show that there is a significant positive effect of paternal income on offspring income of Chinese residents, and every 1% increase in paternal income level will increase the offspring income by 0.265–0.43%. The Intergenerational Income Elasticity of our residents from 2010–2018 ranged from about 0.265–0.430. Using data from Chinese Household Income Project Survey (CHIP), Chen Lin estimates that the IGE in China is 0.4 for the period 1998–2002 [25]. Yang and Shi estimate that the IGE in China is about 0.339, based on data from the China Family Panel Studies (CFPS) in 2010 [23]. Yang and Wang use data from the China Health and Nutrition Survey (CHNS) to measure the IGE values of 0.315 and 0.266 for Chinese residents in 2011 and 2015, respectively [24]. Differences in databases, years, and estimation methods may cause differences in the results of IGE estimates. However, in general, the estimation of IGE in this paper is basically consistent with the measurement results of scholars, and it also confirms the existence of obvious intergenerational correlations of residents' income in China.

**Table 1. Intergenerational income elasticity in China (2010–2018).**

| Variable | $lnY_1$ | |
| --- | --- | --- |
| | **(1) OLS** | **(2) 2SLS** |
| $lnY_0$ | 0.265*** (27.49) | 0.430*** (17.20) |
| $age_1$ | 0.213*** (18.00) | 0.211*** (20.53) |
| $age_0$ | 0.086*** (5.59) | 0.077*** (5.40) |
| $age_1^2$ | -0.003*** (-13.95) | -0.003*** (-16.10) |
| $age_0^2$ | -0.001*** (-6.01) | -0.001*** (-5.41) |
| constant | 1.238*** (3.48) | -0.259 (-0.67) |
| N | 10356 | 10356 |
| $R^2$ | 0.200 | 0.172 |
| F | 438.408 | 385.320 |
| Weak identification test (F- statistic) | | 762.702 |
| Sargan statistic (P-value) | | 0.0625 |

*Notes*: $t$ statistics in parentheses,

\*$p < 0.1$;

\*\*$p < 0.05$;

\*\*\*$p < 0.01$.

## 5.2 The effect of intergenerational income mobility on MPC

**5.2.1 Benchmark results.** To analyze the effect of intergenerational income mobility on MPC of China's residents, we group the 10,356 samples according to province ID and year, and then estimate the IGE and MPC of the parent generation ($MPC_1$ and $MPC_2$) for each group separately, obtaining 125 groups of provincial panel data. Then we test the effect of the level of intergenerational income mobility on MPC of the paternal generation according to Eq (13). The results are shown in Tables 2 and 3.

In Tables 2 and 3, column (1) shows the results of random effects, column (2) shows the results of individual fixed effects, and column (3) shows the results of individual and time dual fixed effects. The results of the Hausmann test in Tables 2 and 3 were 0.1124 and 0.2160, respectively. The random effects should be chosen for this panel data, so we use the estimation results in column (1) and the fixed effects results as a reference. The results of the sensitivity analysis given in Table 4 also confirm the validity of the model.

The results in Table 2 show that for every 0.1 increase in the IGE, the parent's marginal propensity to consume ($MPC_1$) increases by 0.371, and the IGE has a positive effect on the residents' marginal propensity to consume at the 1% level of significance. This result is generally consistent with the inference of Hypothesis 1. Table 3 shows that for every 0.1 increase in the IGE, the residents' marginal propensity to consume education ($MPC_2$) will increase by 0.203. Combined with Table 2, it is easy to conclude that for every 0.1 increase in the IGE, 37.1% of a unit increase in residents' income will be spent on increased consumption, of which 20.3% will be spent on education expenditures and 16.8% on consumption expenditures other than education. Currently, spending on culture, education and entertainment accounts for about 10% of the per capita consumption expenditure of Chinese residents. Thus, when the Intertemporal Income Elasticity increases, more than half of the increased consumption of residents is spent on education, which will increase the share of education expenditure in the consumption of residents. Combined with the fact that household education expenditure in China is mainly spent by parents on their children's education, it can be concluded that " The higher the IGE,

**Table 2. Effect of intergenerational income elasticity on $MPC_1$.**

| Variable | | MPC_1 | | |
|---|---|---|---|---|
| | | **(1)** | **(2)** | **(3)** |
| Core explanatory variable | IGE | 3.706*** (4.48) | 3.087*** (3.60) | 3.008*** (3.35) |
| Control Variables | $lnY_0$ | 0.783* (1.92) | 1.137** (2.56) | 1.602** (2.42) |
| | $lnasset_0$ | -1.631*** (-3.00) | -2.624*** (-3.88) | -2.282*** (-3.26) |
| | $y$ | 0.141 (0.89) | -0.097 (-0.38) | -0.251 (-0.77) |
| | gini | -1.082 (-0.25) | 4.278 (0.90) | 2.284 (0.45) |
| | $age_0$ | -0.541** (-2.16) | -0.931*** (-2.67) | -1.080*** (-2.84) |
| | $age_1$ | 0.674** (2.15) | 1.181** (2.62) | 1.321*** (2.73) |
| | popu | 0.058 (1.04) | 0.192 (1.56) | 0.136 (0.91) |
| | urb | 11.927*** (3.00) | -4.239 (-0.28) | -6.792 (-0.36) |
| | $edu_0$ | -0.104 (-1.11) | -0.088 (-0.82) | -0.768** (-2.16) |
| | pattern | -0.111 (-0.72) | -0.094 (-0.53) | -0.077 (-0.43) |
| Individual FE | | N | Y | Y |
| Time FE | | N | N | Y |
| constant | | 13.993* (1.67) | 34.937** (2.23) | 41.001** (2.03) |
| N | | 125 | 125 | 125 |
| $R^2$ | | | 0.105 | 0.107 |
| F | | | 4.506 | 3.594 |

*Notes*: *t* statistics in parentheses,

*$p < 0.1$;

**$p < 0.05$;

***$p < 0.01$.

the stronger the altruistic motive of residents' consumption", which confirms the inference of Hypothesis 2.

**5.2.2 Robustness check.** In order to ensure the reliability of the above results, we adopt the method of replacing the core explanatory variable indicator to do the robustness check. The core explanatory variable indicator was replaced with "intergenerational income rank correlation coefficient (IGC)" (Wang et al., 2019), and then Eq (13) was estimated using $MPC_1$ and $MPC_2$ as explained variables, respectively. The results are shown in Table 5.

In Table 5, columns (1) and (3) show the results of random effects, columns (2) and (4) show the results of individual fixed effects. The estimation results in Table 5 are strongly consistent with Tables 2 and 3: the stronger the degree of intergenerational income association, the greater the residents' marginal propensity to consume; and a larger share of the increased consumption expenditure is spent on education, leading to an increasing altruistic motive for residents' consumption.

**5.2.3 Relative importance analysis.** On the basis of the benchmark model regression, we use the Shapley value decomposition method to decompose the marginal propensity to consume and perform a relative importance analysis of the independent variables in order to measure the degree of contribution of the respective variables in explaining the MPC. In Table 6, the top five important influences on $MPC_1$ and $MPC_2$ and their contribution are presented separately.

The results in Table 6 show that the contribution of IGE in explaining $MPC_1$ is about 9%, which is the fourth most important influence on $MPC_1$, after the net household property, the urbanization rate and the household income. The contribution of IGE in explaining $MPC_2$ is

**Table 3. Effect of intergenerational income elasticity on $MPC_2$.**

| Variable | | MPC$_2$ | | |
|---|---|---|---|---|
| | | **(1)** | **(2)** | **(3)** |
| Core explanatory variable | IGE | 2.032*** (5.22) | 1.778*** (4.28) | 1.562*** (2.86) |
| Control Variables | $lnY_0$ | 1.196*** (6.23) | 1.291*** (6.01) | -0.351 (-0.87) |
| | $lnasset_0$ | -0.710*** (-2.74) | -0.956*** (-2.92) | -1.426*** (-3.35) |
| | $y$ | 0.014 (0.19) | -0.122 (-0.99) | -0.199 (-1.01) |
| | gini | -0.270 (-0.13) | 1.224 (0.53) | 3.025 (0.98) |
| | $age_0$ | -0.193 (-1.61) | -0.321* (-1.90) | -0.771*** (-3.33) |
| | $age_1$ | 0.211 (1.40) | 0.366* (1.68) | 0.982*** (3.33) |
| | popu | -0.002 (-0.07) | 0.023 (0.38) | 0.089 (0.98) |
| | urb | 1.745 (0.91) | -6.630 (-0.91) | -7.974 (-0.69) |
| | $edu_0$ | -0.043 (-0.98) | -0.012 (-0.22) | -0.669*** (-3.09) |
| | pattern | -0.029 (-0.39) | -0.027 (-0.31) | -0.035 (-0.32) |
| Individual FE | | N | Y | Y |
| Time FE | | N | N | Y |
| constant | | -0.247 (-0.06) | 9.185 (1.21) | 45.944*** (3.73) |
| N | | 125 | 125 | 125 |
| $R^2$ | | | 0.272 | 0.157 |
| F | | | 7.389 | 4.143 |

*Notes*: *t* statistics in parentheses,

*$p < 0.1$;

**$p < 0.05$;

***$p < 0.01$.

about 14.5%, which is the third most important influence on $MPC_2$, after the household income and the net household property.

Both theoretical and empirical analyses show that generational factors play a significant role in the marginal propensity to consume in China. A decline in intergenerational income mobility leads to a higher marginal propensity to consume. The main logic behind this is that the father's generation, for intergenerational reasons, will skew consumer spending toward paying for the education of the offspring. On the surface, weakened intergenerational income mobility could not only stimulate residents' willingness to consume, but also increase the share of developmental consumption in total consumption. However, under this "altruistic motive", the "improvement" of residents' willingness to consume and consumption structure cannot really meet people's diversified needs for a better life, nor can it really improve people's living standard. Most of the residents' stimulated willingness to consume and increased consumption expenditure are paid for their offspring, and the improvement of their own quality of life and their own value is not synchronized with the growth of their income and

**Table 4. Sensitivity analysis.**

| | MPC$_1$ | | MPC$_2$ | |
|---|---|---|---|---|
| | **Coef.** | **P value** | **Coef.** | **P value** |
| betaxL | 3.614 | 0.001 | 1.907 | 0.000 |
| betaxH | 3.706 | 0.001 | 2.032 | 0.000 |
| betax (Conservative confidence interval) | 2.61 | 7.07 | 1.241 | 4.414 |

**Table 5. Robustness check.**

| Variable | | MPC$_1$ | | MPC$_2$ | |
|---|---|---|---|---|---|
| | | **(1) RE** | **(2) FE** | **(3) RE** | **(4) FE** |
| Core explanatory variable | IGC | 3.843*** (5.82) | 3.648*** (5.09) | 2.135*** (4.95) | 2.066*** (4.24) |
| Control Variables | $lnY_0$ | 0.921** (2.41) | 1.144*** (2.78) | 1.296*** (6.81) | 1.349*** (6.35) |
| | $lnasset_0$ | -1.461*** (-2.79) | -2.211*** (-3.41) | -0.680** (-2.53) | -0.833**(-2.50) |
| | y | 0.234 (1.54) | 0.063 (0.26) | 0.053 (0.68) | -0.065 (-0.52) |
| | gini | -4.873 (-1.16) | -0.463 (-0.10) | -1.749 (-0.83) | -0.732 (-0.31) |
| | $age_0$ | -0.451* (-1.88) | -0.735** (-2.21) | -0.151 (-1.21) | -0.242 (-1.41) |
| | $age_1$ | 0.346 (1.12) | 0.701 (1.60) | 0.078 (0.48) | 0.178 (0.79) |
| | popu | 0.040 (0.76) | 0.245** (2.11) | -0.010 (-0.36) | 0.045 (0.75) |
| | urb | 11.949*** (3.12) | -6.175 (-0.44) | 1.577 (0.77) | -8.489 (-1.17) |
| | $edu_0$ | -0.067 (-0.75) | -0.076 (-0.75) | -0.025 (-0.57) | -0.003 (-0.06) |
| | pattern | -0.058 (-0.39) | -0.012 (-0.07) | 0.002 (0.02) | 0.013 (0.16) |
| constant | | 14.951* (1.85) | 30.954** (2.09) | -0.026 (-0.01) | 8.077 (1.06) |
| N | | 125 | 125 | 125 | 125 |
| $R^2$ | | | 0.206 | | 0.270 |
| F | | | 6.111 | | 7.346 |

*Notes*: *t* statistics in parentheses,

*$p < 0.1$;

**$p < 0.05$;

***$p < 0.01$.

consumption. In the face of strong budget constraints, increased altruistic motivation is sometimes accompanied by a degree of "self-sacrifice". In other words, under the influence of intergenerational factors, the father may reduce his own consumption in order to pay for the education of his offspring, so that altruistic consumption may crowd out "selfish consumption", resulting in the coexistence of "altruistic consumption upgrading" and "selfish consumption downgrading". Altruistic consumption is not permanent. The altruistic motivation for consumption will be significantly weaker when the offspring finish their education. However, at this time, the "self-interested consumption" of the father's generation is difficult to upgrade under the effect of inertia, and the propensity to consume decreases significantly. This "side effect" will become more pronounced as the population ages.

Therefore, it is necessary to consider the "motivation" behind consumption to promote the expansion and upgrading of consumerism. Reducing the "altruistic motive" will help unleash people's consumption potential and better serve the economy's internal circulation and high-quality development.

**Table 6. Relative importance of factors on marginal propensity to consume.**

| Variable | MPC$_1$ | | Variable | MPC$_2$ | |
|---|---|---|---|---|---|
| | **Contribution (%)** | **Rank** | | **Contribution (%)** | **Rank** |
| $asset_0$ | 47.803 | 1 | $Y_0$ | 46.819 | 1 |
| urb | 12.367 | 2 | $asset_0$ | 29.086 | 2 |
| $Y_0$ | 10.032 | 3 | IGE | 14.517 | 3 |
| IGE | 9.309 | 4 | $age_1$ | 2.617 | 4 |
| $age_1$ | 6.732 | 5 | popu | 1.882 | 5 |

**5.2.4 Analysis of urban-rural heterogeneity.** The urban-rural differences that exist in our country in terms of social, cultural and economic aspects. Urban and rural residents may also show differences in income levels, consumption propensity and consumption characteristics. These differences affect the role of the level of intergenerational income mobility on the propensity to consume and the "altruistic motive" of consumption. This section of the paper discusses the heterogeneity of the impact of the level of intergenerational income mobility on the propensity to consume from an urban-rural perspective.

The previous section has used provincial panel data to analyze the effect of intergenerational income elasticity on residents' marginal propensity to consume and marginal propensity to consume education. The sample size was 125. If the sample is then grouped into urban and rural areas, the small sample size in each group will increase the bias of the estimation results. Therefore, the paper performs the heterogeneity analysis by constructing interaction terms. We introduce the interaction term of Intergenerational Income Elasticity (IGE) and urban-rural classification (urban) into the model, and then estimate the model with $MPC_1$ and $MPC_2$ as explanatory variables, respectively. The results are shown in Table 7.

The value of Hausman test in columns (1) and (2) of Table 7 is 0.0020, and the value of the test in columns (3) and (4) is 0.0220, respectively. The results indicate that the fixed-effects model should be chosen, so we use the estimation results in columns (2) and (4), and the random-effects estimation results in columns (1) and (3) are for reference. The results in Table 7 show that the interaction term has a positive effect on $MPC_1$ and $MPC_2$ at the 1% level of significance. Combining the results in Tables 2 and 3 again, it can be judged that the urban-rural classification (urban) reinforces the effect of Intergenerational Income Elasticity on residents' marginal propensity to consume and marginal propensity to consume education. This

**Table 7. Analysis of urban-rural heterogeneity.**

| Variable | $MPC_1$ | | $MPC_2$ | |
|---|---|---|---|---|
| | **(1) RE** | **(2) FE** | **(3) RE** | **(4) FE** |
| *IGE#urban* | 5.000*** (2.89) | 3.965** (2.06) | 2.513*** (3.08) | 2.608*** (2.86) |
| *IGE* | 0.486 (0.36) | 0.593 (0.40) | 0.412 (0.64) | 0.138 (0.20) |
| $lnY_0$ | 0.454 (1.11) | 0.833* (1.81) | 1.030*** (5.34) | 1.092*** (5.01) |
| $lnasset_0$ | -1.376** (-2.56) | -2.234*** (-3.24) | -0.578** (-2.24) | -0.700** (-2.13) |
| *y* | 0.230 (1.47) | -0.040 (-0.16) | 0.056 (0.74) | -0.085 (-0.71) |
| *gini* | -1.395 (-0.33) | 3.241 (0.69) | -0.414 (-0.21) | 0.541 (0.24) |
| $age_0$ | -0.422* (-1.70) | -0.743** (-2.09) | -0.129 (-1.07) | -0.197 (-1.17) |
| *age* | 0.570* (1.85) | 0.974** (2.15) | 0.155 (1.03) | 0.230 (1.07) |
| *popu* | 0.070 (1.29) | 0.220* (1.81) | 0.003 (0.12) | 0.041 (0.71) |
| *urb* | 12.489*** (3.21) | -7.238 (-0.49) | 1.928 (1.01) | -8.602 (-1.22) |
| $edu_0$ | -0.127 (-1.40) | -0.102 (-0.96) | -0.053 (-1.25) | -0.021 (-0.41) |
| *pattern* | -0.036 (-0.24) | -0.020 (-0.11) | 0.010 (0.14) | 0.022 (0.26) |
| *constant* | 9.231 (1.10) | 29.353* (1.88) | -2.686 (-0.66) | 5.512 (0.74) |
| *N* | 125 | 125 | 125 | 125 |
| $R^2$ | | 0.137 | | 0.326 |
| *F* | | 4.637 | | 8.003 |

*Notes*: *t* statistics in parentheses,

*$p < 0.1$;

**$p < 0.05$;

***$p < 0.01$.

confirms Hypotheses 3 and 4: Intergenerational Income Elasticity has a stronger positive effect on urban residents' marginal propensity to consume and marginal propensity to consume education than rural residents. Under the effect of intergenerational factors, urban residents' consumption will show stronger altruistic motives.

# 6 Research conclusions and recommendations

## 6.1 Conclusions

This paper analyzes the theoretical mechanism through which intergenerational income mobility affects residents' marginal propensity to consume and altruistic consumption motives by extending the Intergenerational Overlap Model. The CFPS data from 2010–2018 are used to test the theoretical mechanism and analyze the degree of influence and contribution of intergenerational income mobility on the propensity to consume of Chinese residents. The conclusions of the paper are as follows:

The level of intergenerational income mobility is an important factor affecting the marginal propensity to consume and the marginal propensity to consume in education. Intergenerational Income Elasticity has a significant positive effect on residents' marginal propensity to consume. When Intergenerational Income Elasticity increases, the "altruistic motive" of residents' consumption will also increase significantly.

There is urban-rural heterogeneity in the effect of intergenerational income mobility on residents' marginal propensity to consume and "altruistic motives". Compared with rural residents, Intergenerational Income Elasticity has a stronger positive effect on urban residents' marginal propensity to consume and marginal propensity to consume education. With the effect of intergenerational factors, urban residents' consumption shows stronger altruistic motives.

## 6.2 Recommendations

There is a positive effect of Intergenerational Income Elasticity on the altruistic motive of consumption. Considered from an intergenerational perspective, increasing the level of intergenerational income mobility can release the potential of consumption and diminish the altruistic motive of consumption.

To build an open education system and reduce the burden of education expenditure on residents. We can make full use of the Internet platform and digital information technology to guide the free supply of quality educational resources at different levels, including basic education, higher education and vocational education. Improve the attributes of public goods in education and build an open education system. An open education system can help reduce the burden of education expenditure on residents, alleviate the squeeze of education expenditure on other consumption expenditure, reduce the "altruistic motive" of residents' consumption, and release the potential of consumption. An open education system helps to narrow the gap between urban and rural education and improve the equity of education and the level of human capital in society. The overall improvement in the level of social human capital helps optimize and upgrade the consumption structure.

Improve the tax system to reduce the amount of intergenerational transfer of wealth. Intergenerational wealth transfer is an important factor affecting the level of intergenerational income mobility. Improving the tax system and reducing the amount of intergenerational wealth transfers will help increase the intergenerational income mobility of society. We should continuously improve the supply of tax systems such as inheritance tax and gift tax, and promote the implementation of the three-dimensional distribution of national income by means of taxation. By reducing the amount of intergenerational transfer of wealth and improving the

inequality of income distribution, we can enhance the consumption level and consumption willingness of residents.

Reduce the pressure of intergenerational support by improving social security mechanisms. Altruistic consumption is not only about education expenses. Intergenerational spending on retirement and health care is also considered altruistic consumption. Improving social security mechanisms can help reduce the "altruistic motive" of consumption. Utilize the functions of the government and the market to establish a multi-input mechanism to improve the quantity and quality of the supply of medical and elderly services, and to reduce the price of the supply of these services. Relying on government power to eliminate restrictions on medical and pension insurance recipients by the household registration factor, expand the coverage of basic medical and pension insurance, and gradually eliminate urban-rural and regional disparities in social security provision.

## Author Contributions

**Data curation:** Yingying Shao.

**Formal analysis:** Yingying Shao.

**Methodology:** Yingying Shao.

**Writing – original draft:** Yingying Shao.

**Writing – review & editing:** Yingying Shao.

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
