## [Decision Letter · Decision Letter 0]

21 Aug 2023

PONE-D-23-19313A Study on Altruistic Consumption of Chinese Residents from the Perspective of Intergenerational Income MobilityPLOS ONE

Dear Dr. SHAO,

Thank you for submitting your manuscript to PLOS ONE. After careful consideration, we feel that it has merit but does not fully meet PLOS ONE’s publication criteria as it currently stands. Therefore, we invite you to submit a revised version of the manuscript that addresses the points raised during the review process.

We look forward to receiving your revised manuscript.

Kind regards,

Wajid Khan

Academic Editor

PLOS ONE

Reviewers' comments:

Reviewer's Responses to Questions

**Comments to the Author**

1. Is the manuscript technically sound, and do the data support the conclusions?

Reviewer #1: Yes

Reviewer #2: Yes

2. Has the statistical analysis been performed appropriately and rigorously? 

Reviewer #1: Yes

Reviewer #2: Yes

3. Have the authors made all data underlying the findings in their manuscript fully available?

Reviewer #1: Yes

Reviewer #2: Yes

4. Is the manuscript presented in an intelligible fashion and written in standard English?

Reviewer #1: Yes

Reviewer #2: Yes

5. Review Comments to the Author

Reviewer #1: This paper expands the Over Lapping Generation Models on the Chinese family panel study data for an analysis of intergenerational income mobility. A wide evaluation verifies the significant effect of the integrational income elasticity on urban residents’ marginal propensity to consume. The positive relation of this metric with the altruistic motive of residents’ consumption is also proved via an empirical investigation.

Generally speaking, the paper is clearly written and reasonably organized. I have the following concerns during my reading:

1) The research in this paper is promoted based on the Diamond’s Overlapping Generation model, yet this baseline has not been formally introduced and cited its reference. Better to describe this baseline in the Related Work of Preliminary section.

2) What is the division standard of youth, middle aged, and old aged residents? Better to specify this standard.

3) The meanings of some notations have not been well explained, such as a_0, a_1, and a_2 in Eqs. (12) and (13), \\varepsilon_i in Eq. (12), and \\delta_i in Eq. (13).

4) Some discussions lack theoretical supports from existing studies. For example, it’s argued that “7%-9% of the income composition of China’s residents is property income, and over 70% is wage and business income”. This discussion has not been pointed out the literature where it comes from. Better to properly review existing studies and cite them in your paper;

5) There lacks an evaluation regarding parameter effects, such as the interest rate in Eq. (3) and \\alpha_1 and \\laphg_2 in Eqs. (10) and (11). Better to test the parameter sensitivity of the model for a convincing evaluation.

6) It seems that there lacks an experimental evidence to validate the discussion that the intergenerational income elasticity has a stronger effect on urban residents' marginal propensity to consume than on the rural residents’ propensity. How do you validate this issue, as argued in Abstract?

Reviewer #2: This paper analyzes the theoretical mechanism through which intergenerational income mobility affects residents' marginal propensity to consume and altruistic consumption.

In the field of intergenerational income mobility research, a number of literature reviews have been analyzed for measuring the level of intergenerational income mobility.

The authors perform an analysis based on the OLG model. Also, in the paper, IGE is used to measure the power of the level of intergenerational income mobility. Based on the measurement of IGE and MPC, the authors build a model to analyze the effect of Intergenerational Income Elasticity on residents' Marginal Propensity to Consume using provincial panel data.

The results of the Hausman test required simple processing that led to the validation of the hypotheses assumed by the authors.

In the Introduction section, the values of several studies are highlighted, but the identified reference sources are not mentioned.

The bibliography does not follow alphabetical order.

6. PLOS authors have the option to publish the peer review history of their article (what does this mean?). If published, this will include your full peer review and any attached files.

Reviewer #1: No

Reviewer #2: **Yes: **Valentin Radu

---

## [Author Response · Author response to Decision Letter 0]

30 Sep 2023

First, I would like to thank the academic editors and reviewers for suggesting changes to my manuscript. The following responses are made to each point raised by the academic editor and reviewers.

1) The research in this paper is promoted based on the Diamond’s Overlapping Generation model. However, the Overlapping Generation model was not introduced in the previous manuscript, resulting in the Related Literature section of the paper was incomplete. In the revised manuscript, we have added the OLG model in the Related Literature section. The cited references have also been added.

2) The division standard of youth, middle aged, and old aged residents have been explained in the article. Youth are defined as those who are not yet able to work and are in the education stage. Middle aged residents are defined as those who are able to earn income from labor and need to raise children. And old aged residents are defined as those who are not able to work and therefore cannot continue to earn income from labor. 

3) The reviewers noted that the meanings of some notations have not been well explained such as a0, a1,δi in Eqs. (12) and (13). In the revised manuscript, we have made additions.

4) “In the last decade, about 7%-9% of the income composition of China's residents is property income, and over 70% is wage and business income.” The data involved in the above sentence were calculated by the author on the basis of official data published by the National Bureau of Statistics of China. The sources of the data have been explained in the revised manuscript.

5) In the Analysis of Urban-Rural Heterogeneity section, we introduced the interaction term of Intergenerational Income Elasticity (IGE) and urban-rural classification (urban) into the model, and then estimate the model with MPC1 and MPC2 as explanatory variables, respectively. The results show that the interaction term has a positive effect on MPC1 and MPC2 at the 1% level of significance. We therefore conclude that: compared with rural residents, Intergenerational Income Elasticity has a stronger positive effect on urban residents' marginal propensity to consume。

6) In the revised manuscript, we have explained the reference sources of the relevant studies and data in the Introduction section. The results of the Hausmann test are also shown in the revised manuscript.

7) The References in the original manuscript were sorted according to the order of citation. Based on the reviewers' suggestions, we reordered the bibliography in alphabetical order in the revised manuscript.

---

## [Decision Letter · Decision Letter 1]

18 Oct 2023

PONE-D-23-19313R1A Study on Altruistic Consumption of Chinese Residents from the Perspective of Intergenerational Income MobilityPLOS ONE

Dear Dr. SHAO,

Thank you for submitting your manuscript to PLOS ONE. After careful consideration, we feel that it has merit but does not fully meet PLOS ONE’s publication criteria as it currently stands. Therefore, we invite you to submit a revised version of the manuscript that addresses the points raised during the review process.

Please submit your revised manuscript by Dec 02 2023 11:59PM. If you will need more time than this to complete your revisions, please reply to this message or contact the journal office at plosone@plos.org. Please include the following items when submitting your revised manuscript:A rebuttal letter that responds to each point raised by the academic editor and reviewer(s). You should upload this letter as a separate file labeled 'Response to Reviewers'.A marked-up copy of your manuscript that highlights changes made to the original version. You should upload this as a separate file labeled 'Revised Manuscript with Track Changes'.An unmarked version of your revised paper without tracked changes. You should upload this as a separate file labeled 'Manuscript'.If applicable, we recommend that you deposit your laboratory protocols in protocols.io to enhance the reproducibility of your results. Protocols.io assigns your protocol its own identifier (DOI) so that it can be cited independently in the future. For instructions see: https://journals.plos.org/plosone/s/submission-guidelines#loc-laboratory-protocols. Additionally, PLOS ONE offers an option for publishing peer-reviewed Lab Protocol articles, which describe protocols hosted on protocols.io. Read more information on sharing protocols at https://plos.org/protocols?utm_medium=editorial-email&utm_source=authorletters&utm_campaign=protocols.

We look forward to receiving your revised manuscript.

Kind regards,

Wajid Khan

Academic Editor

PLOS ONE

Journal Requirements:

Reviewers' comments:

Reviewer's Responses to Questions

**Comments to the Author**

1. If the authors have adequately addressed your comments raised in a previous round of review and you feel that this manuscript is now acceptable for publication, you may indicate that here to bypass the “Comments to the Author” section, enter your conflict of interest statement in the “Confidential to Editor” section, and submit your "Accept" recommendation.

Reviewer #1: (No Response)

Reviewer #2: All comments have been addressed

2. Is the manuscript technically sound, and do the data support the conclusions?

Reviewer #1: Yes

Reviewer #2: Yes

3. Has the statistical analysis been performed appropriately and rigorously? 

Reviewer #1: Yes

Reviewer #2: Yes

4. Have the authors made all data underlying the findings in their manuscript fully available?

Reviewer #1: Yes

Reviewer #2: Yes

5. Is the manuscript presented in an intelligible fashion and written in standard English?

Reviewer #1: Yes

Reviewer #2: Yes

6. Review Comments to the Author

Reviewer #1: Thanks for the revisions. Most of my concerns have been fixed, except the parameter sensitivity evaluation of the model. This evaluation is crucial for determining the prediction performance of the proposed model, as well as making my judgement for the contribution of this study. Please revise this issue and strengthen the soundness of the manuscript. Additionally, the description of the age division standard should be more clearly; I cannot find it in your revised paper.

Reviewer #2: Authors still must improve some aspects that can lead to a better form of paper.

1. Keyword review is required: Marginal propensity to consume / Intergenerational income mobility / Altruistic motivation.

2. In the Introduction section, the authors still start with a numerical presentation without indicating the source. An appropriate wording would be ... According to the source, in 2014, China's total final consumption …

3. At the end of the introduction section, the structure of the work by sections is not presented.

4. In Related Literature Section, Heading 2 are required for Diamond’s Overlapping Generation model (2.1) and Literature on Intergenerational Income Mobility (2.2). Also Heading 3 for Section 4.1 and 5.2 (4.1.1. … ; 5.2.1 ….). Use the same format for headings.

5. In explanation, when you refer to the table columns use "and" (2)(4) use (2) and (4). I also found an error in the paragraph about two words about the column: The results indicate that the fixed-effects model should be chosen, so we use the estimation results in column columns (2)(4) …

7. PLOS authors have the option to publish the peer review history of their article (what does this mean?). If published, this will include your full peer review and any attached files.

Reviewer #1: No

Reviewer #2: **Yes: **Valentin Radu

---

## [Author Response · Author response to Decision Letter 1]

14 Nov 2023

First, I would like to thank the academic editors and reviewers for suggesting changes to my manuscript. The following responses are made to each point raised by the academic editor and reviewers.

1) According to the reviewer's suggestion, I did the sensitivity evaluation of the model in order to show prediction performance of the proposed model. The results of the analysis are presented after the benchmark results. I have also added a description of the age division standard in the paper.

2) I did a review of the keywords and revised them. 

3) In the Introduction section,. the sources of the data are indicated. At the end of the introduction section, I present the structure of the paper by sections.

4) The serial numbers of the article section headings have been added and adjusted. Word errors raised by reviewers have also been corrected.

Thanks again to the academic editors and reviewers for their comments on this paper!

---

## [Decision Letter · Decision Letter 2]

5 Jan 2024

A Study on Altruistic Consumption of Chinese Residents from the Perspective of Intergenerational Income Mobility

PONE-D-23-19313R2

Dear Dr. Shao,

We’re pleased to inform you that your manuscript has been judged scientifically suitable for publication and will be formally accepted for publication once it meets all outstanding technical requirements.

Kind regards,

Wajid Khan

Academic Editor

PLOS ONE

Additional Editor Comments (optional):

Reviewers' comments:

Reviewer's Responses to Questions

**Comments to the Author**

1. If the authors have adequately addressed your comments raised in a previous round of review and you feel that this manuscript is now acceptable for publication, you may indicate that here to bypass the “Comments to the Author” section, enter your conflict of interest statement in the “Confidential to Editor” section, and submit your "Accept" recommendation.

Reviewer #1: All comments have been addressed

Reviewer #2: All comments have been addressed

2. Is the manuscript technically sound, and do the data support the conclusions?

Reviewer #1: Yes

Reviewer #2: Yes

3. Has the statistical analysis been performed appropriately and rigorously? 

Reviewer #1: Yes

Reviewer #2: Yes

4. Have the authors made all data underlying the findings in their manuscript fully available?

Reviewer #1: Yes

Reviewer #2: Yes

5. Is the manuscript presented in an intelligible fashion and written in standard English?

Reviewer #1: Yes

Reviewer #2: Yes

6. Review Comments to the Author

Reviewer #1: Thanks for the revisions. All of my concerns have been fixed, and I’d like to recommend acceptance of this paper.

Reviewer #2: The article has an improved form and the authors took into account the comments of the reviewers. I think it can be published in its current form.

7. PLOS authors have the option to publish the peer review history of their article (what does this mean?). If published, this will include your full peer review and any attached files.

Reviewer #1: No

Reviewer #2: **Yes: **Valentin Radu

---

## [Editor Report · Acceptance letter]

17 Jan 2024

PONE-D-23-19313R2 

PLOS ONE

Dear Dr. Shao, 

I'm pleased to inform you that your manuscript has been deemed suitable for publication in PLOS ONE. Congratulations! Your manuscript is now being handed over to our production team.

Kind regards, 

on behalf of

Dr. Wajid Khan 

Academic Editor

PLOS ONE